# Feline Vector-Borne Diseases and Their Possible Association with Hematological Abnormalities in Cats from Midwestern Brazil

**DOI:** 10.3390/microorganisms12112171

**Published:** 2024-10-29

**Authors:** Stephani Félix Carvalho, Gracielle Teles Pádua, Warley Vieira de Freitas Paula, Mariana Avelar Tavares, Lucianne Cardoso Neves, Brenda Gomes Pereira, Rayane Almeida Santos, Gabriel Cândido dos Santos, Ennya Rafaella Neves Cardoso, Andriele Ferreira Qualhato, Raphaela Bueno Mendes Bittencourt, Nicolas Jalowitzki de Lima, Danieli Brolo Martins, Filipe Dantas-Torres, Felipe da Silva Krawczak

**Affiliations:** 1Laboratório de Doenças Parasitárias (LADOPAR), Setor de Medicina Veterinária Preventiva, Escola de Veterinária e Zootecnia, Universidade Federal de Goiás—UFG, Goiânia 74690-900, GO, Brazil; mv.stephanicarvalho@gmail.com (S.F.C.); gracielletelespadua@discente.ufg.br (G.T.P.); warleyvieira@discente.ufg.br (W.V.d.F.P.); mariana.tavares@discente.ufg.br (M.A.T.); luciannecardoso@discente.ufg.br (L.C.N.); brendagomesp.medvet@gmail.com (B.G.P.); rayalmeida@discente.ufg.br (R.A.S.); doscandido@discente.ufg.br (G.C.d.S.); ennyaneves@discente.ufg.br (E.R.N.C.); rafabmbitt@discente.ufg.br (R.B.M.B.); jalowitzki@discente.ufg.br (N.J.d.L.); 2Laboratório de Patologia Clínica Veterinária, Escola de Veterinária e Zootecnia, Universidade Federal de Goiás—UFG, Goiânia 74690-900, GO, Brazil; andrielefq@discente.ufg.br (A.F.Q.); danieli@ufg.br (D.B.M.); 3Departamento de Imunologia, Instituto Ageu Magalhães—IAM, Fundação Oswaldo Cruz (Fiocruz), Recife 50740-465, PE, Brazil; filipe.torres@fiocruz.br

**Keywords:** PCR, hematology, cats, bloodborne pathogens, vector-borne diseases

## Abstract

Among the parasitic and infectious diseases affecting cats, those caused by vector-borne pathogens deserve attention due to their ability to cause nonspecific clinical signs and clinicopathological abnormalities. We studied the presence of *Cytauxzoon* spp., *Ehrlichia* spp., and *Mycoplasma* spp. in blood samples from 135 cats referred to the veterinary teaching hospital of the Federal University of Goiás in midwestern Brazil. We also investigated co-infections with Feline Immunodeficiency Virus (FIV) and Feline Leukemia Virus (FeLV) as well as the correlation between *Mycoplasma* spp. infection and cat variables, including age, sex, breed, and complete blood count abnormalities. Upon PCR testing, 20.7% (28/135) of samples were positive for *Mycoplasma* spp., 1.5% (2/135) for *Cytauxzoon* spp., and none for *Ehrlichia* spp. Co-infections with *Mycoplasma* spp. and *Cytauxzoon* spp. were detected in the two cats with the latter infection. *Mycoplasma* spp. infection was statistically associated with the simultaneous presence of thrombocytopenia and leukocytosis. This study confirms a high frequence of *Mycoplasma* spp. infection, with both *M. haemofelis* and ‘*Candidatus* Mycoplasma haemominutum’ circulating in this cat population. The clinical significance of *Mycoplasma* spp. infection in cats should be further explored and this infection should eventually be included in the differential diagnosis of thrombocytopenia and leukocytosis in otherwise apparently healthy cats.

## 1. Introduction

Feline vector-borne diseases (FVBDs) affect cats worldwide but are often underestimated as most of the infected cats present a subclinical infection. When present, clinical signs and clinicopathological abnormalities of FVBDs may be nonspecific, thus making it difficult for veterinary practitioners to diagnose this group of diseases [1,2,3]. Cats are more frequently affected by fleas than ticks, sometimes participating in the transmission cycle of flea-borne bacteria of zoonotic concern [4,5].

One of most frequent FVBDs is mycoplasmosis, a disease caused by pleomorphic bacteria that locate on the surface of erythrocytes [6], sometimes inducing hemolytic anemia and fever [7,8]. The prevalence of *Mycoplasma* spp. infection varies widely and co-infections with other vector-borne agents (e.g., *Bartonella henselae, Cytauxzoon felis*, *Leishmania infantum*) have been reported [2,9,10]. Recent studies indicate that *Mycoplasma* spp. are transmitted by multiple pathways, including through vectors, blood transfusion, and aggressive interactions [11].

Domestic cats are mainly infected by three species of hemotropic mycoplasmas (*Mycoplasma haemofelis*, ‘*Candidatus* Mycoplasma haemominutum’, and ‘*Candidatus* Mycoplasma turicensis’), which may cause different clinical signs in infected cats, but are often associated with subclinical infection [7,12]. These three species have been detected in cats in Brazil [13]. Among other vector-borne agents detected in cats worldwide, *Ehrlichia* spp. and *Cytauxzoon* spp. may also be associated with nonspecific clinical signs, making the use of molecular techniques essential for accurate diagnosis and subsequent appropriate treatment [1,14].

FVBDs are endemic in all Brazilian regions, but a relatively low number of studies on *Cytauxzoon*, *Ehrlichia*, and *Mycoplasma* infections in cats have been conducted in midwestern Brazil [15,16,17,18]. Moreover, no study has assessed the association of infection by *Mycoplasma* spp. and complete blood count (CBC) abnormalities in a population of cats. In this perspective, this molecular survey aimed to detect *Cytauxzoon* spp., *Ehrlichia* spp., and *Mycoplasma* spp. in cats referred to a teaching veterinary hospital in midwestern Brazil. We also investigated existing co-infections with Feline Immunodeficiency Virus (FIV) and Feline Leukemia Virus (FeLV) and assessed the correlation between *Mycoplasma* spp. positivity and cat variables, including age, sex, breed, and CBC abnormalities.

## 2. Materials and Methods

### 2.1. Study Area

The study was conducted in the state of Goiás in midwestern Brazil. This region has a tropical climate with two well-defined seasons characterized by a dry period between May and September and a rainy period between October and April. The blood samples from cats included in this study were provided by the veterinary teaching hospital of the Federal University of Goiás. The hospital is located in Goiânia city (16°40′ S, 49°15′ W), Goiás state, but cats came from several municipalities: Goiânia (*n* = 120), Aparecida de Goiânia (*n* = 6), Trindade (*n* = 3), Senador Canedo (*n* = 3), Caldas Novas (*n* = 1), Bonfinópolis (*n* = 1) and Goiatuba (*n* = 1) (Figure 1). Cats were referred to the hospital for routine procedures or other reasons, not necessarily due to suspicion of FVBDs.

### 2.2. Sample Collection

Blood samples were collected (convenience sampling) between January 2022 and December 2023, as part of the routine veterinary care provided by the veterinary teaching hospital. The samples were processed for CBC and then stored at −20 °C until DNA extraction, as described below.

A total of 135 EDTA blood samples (minimum volume, 150 μL) were randomly selected, regardless of the cat’s history and clinical suspicion. Data including municipality of residence, sex, breed, age, and serological results for feline retroviruses (FIV and FeLV) were obtained for each cat included in this study.

The Ethics Committee on Animal Use of the Federal University of Goiás (CEUA/UFG) approved this study (protocol number MB105/23), which was conducted in accordance with the ethical principles of animal experimentation.

### 2.3. CBC Analysis

The CBC analysis included red blood cells (×10^6^/μL), hemoglobin (g/dL), mean corpuscular volume (MCV) (fl), mean corpuscular hemoglobin concentration (MCHC) (%), platelets (×10^3^/μL), and white blood cells (μL), which were measured using an automated cell counter (Celltac Alpha/MEK-6550^®^, Nihon Kohden, Shinjuku City, Japan). A blood smear was also performed on all samples to check the results of the automatic count, to carry out the differential count of white blood cells, to determine the morphology of red blood cells, leukocytes, and platelets, and to analyze the presence of blood pathogens. The packed cell volume (PCV) (%) was determined by the microhematocrit method.

The CBC reference values considered in this study were based on the literature [19] as follows: 5.50–10.00 × 10^6^/μL for red blood cells, 8.0–15.0 g/dL for hemoglobin, 24–45% for hematocrit, 300–800 × 10^3^/μL for platelets, and 5500–19,000/μL for white blood cells.

### 2.4. Immunochromatography for FIV and FeLV

During clinical care, 40% (54/135) of cats were tested for the detection of antibodies to FIV and antigens of FeLV; the assay was performed using the SNAP^®^ ELISA FIV/FELV Combo (IDEXX Laboratories, INC, Westbrook, ME, USA), according to the manufacturer’s guidelines. This ELISA detects FeLV antigens and anti-FIV antibodies. The test was conducted on cats without a history of being tested. The test results were extracted from the electronic medical record of the veterinary teaching hospital of the Federal University of Goiás (HV/EVZ/UFG).

### 2.5. DNA Extraction

DNA was extracted from whole blood samples using the DNeasy Blood and Tissue Kit (QIAGEN, Chatsworth, CA, USA), following the manufacturer’s instructions. After extraction, the DNA samples were stored at −20 °C. The concentration and purity of the obtained DNA samples were assessed using a spectrophotometer (Nanodrop^®^, Thermo Fisher Scientific, Waltham, MA, USA).

### 2.6. PCR Assays

DNA samples were tested by conventional PCR (cPCR) assays for the detection of *Cytauxzoon* spp. and *Ehrlichia* spp. and a real-time PCR (qPCR) for *Mycoplasma* spp. If positive in the qPCR for *Mycoplasma*, the sample was further tested by a cPCR targeting a fragment (≈600 bp) of the *16S rRNA* gene of *Mycoplasma* spp. Details on the target genes, primers, and the expected fragment are reported in Table 1.

cPCR assays were performed using 5 μL of DNA template in a mix containing 12.5 μL of DreamTaq^®^ (Green PCR Master Mix 2X, Thermo Fisher Scientific), 1.5 μL of each primer (forward and reverse) at a concentration of 10 pmol, and 4.5 μL of nuclease-free ultrapure water, totaling a 25 μL reaction volume. The qPCR assays were conducted using the StepOnePlus^TM^ Real-Time PCR System (Applied Biosystems, Foster City, CA, USA), as described by Willi et al. [20]. The 20 μL reaction mixture contained 1.5 μL of DNA template, 10 μL of SYBR Green real-time PCR Master Mix (Applied Biosystems, Thermo Fisher Scientific), 0.5 μL of each primer (forward and reverse) at a concentration of 10 pmol, and 7.5 μL of nuclease-free ultrapure water.

**Table 1 microorganisms-12-02171-t001:** Primers used for the detection of *Cytauxzoon*, *Ehrlichia,* and *Mycoplasma* DNA in cats from Goiás state, midwestern Brazil.

Agents (Target Gene)	Primers	Primer Sequences	Product Size (bp)	References
*Cytauxzoon* spp. (*18S rRNA*) ^b^	Cytaux F	GCGAATCGCATTGCTTTATGCT	284	[21]
	Cytaux R	CCAATTGATACTCCGGAAAGAG		
*Ehrlichia* spp. (*dsb*) ^b^	Dsb-330 (F)	GATGATGTCTGAAGATATGAAACAAAT	409	[22]
	Dsb-728 (R)	CTGCTCGTCTATTTTACTTCTTAAAGT		
*Mycoplasma* spp. (*16S rRNA*) ^a^	SYBR_For	AGCAATRCCATGTGAACGATGAA	134	[20] ^a^
	SYBR_Rev1	TGGCACATAGTTWGCTGTCACTT		
*Mycoplasma* spp. (*16S rRNA*) ^b^	HBT-F	ATACGGCCCATATTCCTACG	595–620	[23] ^b^
	HBT-R	TGCTCCACCACTTGTTCA		

^a^ qPCR; ^b^ cPCR.

No-template controls (PCR-grade water, Sigma-Aldrich, St. Louis, MO, USA) and an appropriate positive control sample (DNA of *Cytauxzoon*, *Ehrlichia* or *Mycoplasma*) were included in each PCR run. Negative samples were further tested using a cPCR assay targeting the cytochrome b gene (*cytB*) of mammals [24,25] to validate the DNA extraction protocol. If a sample did not produce the expected product in this cPCR assay, the sample was excluded from analysis. cPCR products were stained with SYBR Safe (Invitrogen, Thermo Fisher Scientific, Waltham, MA, USA), following the manufacturer’s recommendations, and visualized by electrophoresis in 1.5% agarose gel with an ultraviolet transilluminator.

### 2.7. Partial Mycoplasma spp. 16S rRNA Sequencing and Analysis

Ten PCR amplicons were purified using PureLink™ PCR Micro Kit (Invitrogen, Carlsbad, CA, USA) and prepared for sequencing using BigDye terminator v3.1 (Applied Biosystems, Foster City, CA, USA). Sequencing was conducted in both directions and using the same primers as for PCR, in a 3500xL genetic analyzer (Applied Biosystems, Foster City, CA, USA).

Sequence editing and alignment were conducted using Geneious Prime^®^ 2023.2.1. In brief, low-quality ends from both forward and reverse sequences were trimmed (error probability limit value of 0.05). Then, sequences were verified by eye, edited manually, and aligned using ClustalW (version 2.1). Consensus sequences were compared to sequences available in GenBank using the Basic Local Alignment Search Tool (BLASTn; http://blast.ncbi.nlm.nih.gov/Blast.cgi accessed on 29 September 2024).

*16S rRNA* sequences from hemotropic *Mycoplasma* spp. detected in cats in different countries [12] were exported from GenBank and aligned using ClustalW. The evolutionary history was inferred using the neighbor-joining method [26]. The percentage of replicate trees in which the associated taxa clustered together in the bootstrap test (10,000 replicates) are shown above the branches [27]. The evolutionary distances were computed using the p-distance method [28] and are in the units of the number of base differences per site. This analysis involved 43 sequences. All ambiguous positions were removed for each sequence pair (pairwise deletion option). There were 524 positions in the final dataset. Evolutionary analyses were conducted in MEGA11 [29,30]. *Acholeplasma laidlawii* (accession number: OP501868) and *Mycoplasma pneumoniae* (accession number: M29061) were used as outgroups. The final phylogenetic three was edited using iTOL v.6 (https://itol.embl.de/ accessed on 30 September 2024).

### 2.8. Statistical Analysis

The cat breed and age group distributions were assessed using a One-way Chi-squared test in MedCalc Software Ltd. (https://www.medcalc.org/calc/chisquared-1way.php accessed on 30 September 2024). The association between *Mycoplasma* spp. positivity and other variables (sex, age group, CBC abnormalities, or seropositivity for feline retroviruses) was investigated using a Chi-squared test implemented in QuickCalcs of GraphPad (https://www.graphpad.com/quickcalcs/ accessed on 30 September 2024). A *p*-value < 0.05 was considered statistically significant.

## 3. Results

The cat population investigated herein consisted of 70 (51.9%) males and 65 (48.1%) females. Most cats (*n* = 115) were crossbreeds, and the remaining cats (*n* = 20) belonged to four breeds: Siamese (*n* = 14), Persian (*n* = 4), Angora (*n* = 1), and Bengal (*n* = 1) (One-way Chi-squared test = 362.7407; *df* = 4; *p* < 0.0001). Most of the cats were 1–6 years old (One-way Chi-squared test = 59.9892; *df* = 3; *p* < 0.0001; cats with unknown age excluded from this analysis) (Table 2).

All samples that were negative for *Cytauxzoon* spp., *Ehrlichia* spp., and *Mycoplasma* spp. tested positive in the cPCR assay for the detection of the mammalian endogenous control (*cytB* gene) and, therefore, were included in the analyses. Out of 135 cats tested, 28 (20.7%) were positive for *Mycoplasma* spp., 2 (1.5%) for *Cytauxzoon* spp., and none for *Ehrlichia* spp. The positivity for *Mycoplasma* spp. was significantly higher in males (34.3%) than in females (6.2%) (Two-way Chi-squared test = 16.226, *df* = 1, *p* = 0.0001), but no differences were found in relation to age group (Two-way Chi-squared test = 3.5224, *df* = 3, *p* = 0.3179; cats with unknown age excluded) or breed (Two-way Chi-squared test = 3.2767, *df* = 4, *p* = 0.5126).

Out of 28 cats positive for *Mycoplasma* spp. by qPCR, 27 were positive by cPCR. Ten positive samples with more DNA (larger and brighter positive bands in the agarose gel) were selected for sequencing, but only two partial *16S rRNA* gene sequences were successfully generated, one (491 bp) corresponding to *Mycoplasma haemofelis* (100% query coverage and identity with several sequences in GenBank: MN240855.1, MK632350.1, PP494713.1) and the other (450 bp) to ‘*Candidatus* Mycoplasma haemominutum’ (100% query coverage and identity with several sequences in GenBank: MK632390.1, OQ397117.1, MW598400.1). The low number (only two) of sequences obtained can be attributed to the low number of copies of the target DNA in the tested samples. Our sequences were deposited in GenBank under the accession numbers PQ423636 (*Mycoplasma haemofelis*) and PQ425223 (‘*Candidatus* Mycoplasma haemominutum’). Our sequences clustered with sequences of *M. haemofelis* and ‘*Candidatus* Mycoplasma haemominutum’ previously detected in cats from Brazil and from other countries (Figure 2).

Out of fifty-four cats tested for feline retroviruses, nine (16.7%) were positive for FeLV, one (1.8%) for FIV, and two (3.7%) for both species. Among the *Mycoplasma* spp.-positive cats, four (14.3%) were co-infected with FIV, FeLV, or both (Table 3).

Most of the cats (80.7%) presented at least one CBC abnormality. The results of CBC alterations related to *Mycoplasma* spp. qPCR positivity and FIV/FeLV seropositivity are shown in Table 4. *Mycoplasma* qPCR positivity was statistically associated with the simultaneous presence of thrombocytopenia and leukocytosis (Table 5).

Through the cytological examination of stained blood smears, only two cats (1.5%) were positive for FVBDs showing structures suggestive of *Mycoplasma* spp.; these animals were positive in the qPCR assay for *Mycoplasma* spp.

## 4. Discussion

The prevalence (20.7%) of *Mycoplasma* spp. infection in cats examined herein is consistent with that reported in some studies conducted in Brazil, but it is acknowledged that prevalence data vary widely among studies [13]. Indeed, such comparisons should be made with caution due to the differences in terms of the study populations (e.g., healthy versus sick cats) and diagnostic techniques [13]. For example, our study assessed a cat population that sought veterinary care, and this may have influenced our results. Because we did not evaluate a healthy cat population, our results may not represent the actual prevalence of *Mycoplasma* spp. in the overall cat population in the region. Similarly, the low positivity detected by cytological examination of blood smears highlights its lower sensitivity as compared to PCR and emphasizes the importance of using molecular tools for a reliable diagnosis of *Mycoplasma* spp. infection in cats [15]. In fact, due to cyclic bacteremia and the chronic nature of *Mycoplasma* spp. in cats, cytology is not reliable for diagnosing this infection [13,31].

Although species-specific PCR assays have been developed for *Mycoplasma* spp., their use as screening tools is limited due to the diversity of species [20]. For this reason, we used an SYBR Green-based qPCR assay targeting a fragment of the *16S rRNA* gene of *Mycoplasma* spp., which presents 98.2% sensitivity and 92.1% specificity for feline hemoplasmas [20]. This assay was chosen for initial screening due to its ability to detect several *Mycoplasma* spp., including potentially new species [20].

The higher prevalence of hemotropic mycoplasmas in male cats has been reported in other studies and can partly be explained by their territorial behavior, which results in frequent intraspecific aggressions, thus increasing the possibility of bite-related transmission [11]. Furthermore, *Mycoplasma* spp. infections are more common in stray cats or those that roam freely between the house and the street [13].

Thrombocytopenia and leukocytosis were associated with *Mycoplasma* spp. infection in the present study, although actual species involved in most of these cases could not be determined. Thrombocytopenia is not a feature of *M. haemofelis* infection in cats, but a previous study also found an association between thrombocytopenia and *M. haemofelis* infection in southeastern Brazil [8]. The changes in the leukogram of cats with mycoplasmosis usually do not follow a defined pattern [32]. However, the leukocytosis observed in the present study may be related to the inflammatory process. Leukocytosis was also observed in felines affected by *M. haemofelis*, which showed increased inflammatory biomarkers in response to endothelial glycocalyx damage, a regulator of leukocyte adhesion and migration within the blood vessel [33]. *M. haemofelis* is the most pathogenic *Mycoplasma* sp. in cats and may cause hemolytic anemia, even in immunocompetent cats [8]. The lack of association between anemia and *M. haemofelis* in some studies, including herein, has been attributed to the inclusion of chronically infected asymptomatic cats, considering that anemia is most frequent in the acute phase of the infection [8].

One of the factors that can reduce the frequence of vector-borne pathogens in cats is their grooming behavior that reduces the prevalence and intensity of ectoparasite infestation, especially by ticks [34]. The early removal of ticks by cats may also help prevent pathogen transmission, as ticks need to attach for some time before they can transmit certain pathogens [35]. For instance, the transmission time may vary from 3 to 50 h for *Ehrlichia* spp. [35,36] and from 36 to 48 h for *Cytauxzoon* spp. [37].

Another possible reason for the low positivity of *Cytauxzoon* spp. in our study may be related to the low pathogenicity of the species infecting felids in Brazil [38], as is the case of the recently described *Cytauxzoon brasiliensis* [39]. Indeed, clinical cases of cytauxzoonosis are considered uncommon in this country and it is plausible to infer that cats included in this study were not referred to the hospital due to clinical suspicion of cytauxzoonosis.

Despite the low positivity, this is the first report of *Cytauxzoon* spp. infecting domestic cats in Goiás, midwestern Brazil, where another study detected *Cytauxzoon felis* in free-ranging *Puma concolor* [40]. It is known that cytauxzoonosis has ixodid ticks as definitive hosts and felids as intermediate hosts [16]. For this reason, this is a disease more commonly detected in wild felids. Concerning ixodid ticks, *Amblyomma parvum* was found to parasitize *Leopardus pardalis* infected with *Cytauxzoon* sp. (phylogenetically related to *C. felis*). However, the collected ticks were negative for *Cytauxzoon* spp. [41].

In a recent study, Fagundes-Moreira et al. [42] highlighted that *Amblyomma sculptum* could be a possible vector of *Cytauxzoon* spp., as several ticks of this species were found on infected wild felids in central-western Brazil. The ecological characteristics of *Amblyomma* spp. make their parasitism on domestic cats relatively rare, as they are typically found in forests and pastures. The positive cats found in this study were free of ticks, and their domiciled or semi-domiciled lifestyle makes their infestation by *Amblyomma* spp. ticks unlikely.

The absence of *Ehrlichia* spp. in the studied cats is consistent with other studies conducted in Rio Grande do Sul [43] and São Paulo [17], but some studies have reported the presence of *E. canis* in cats in Brazil [1,35]. In a study carried out in different Brazilian states, infected cats were detected in Minas Gerais and São Paulo, but not in Rondônia [1]. Overall, 2% of the 306 cats tested positive for the *Ehrlichia* spp. *dsb* gene and all sequences were assigned to *E. canis* [1]. Studies conducted in other countries such as Spain, Portugal, and Germany have reported the molecular detection of *Ehrlichia* spp. in 0 to 5.4% of the examined cats [2,44,45,46]. While we did not detect positive cats, Paula et al. [47] conducted a study in the same area investigated herein and reported a seroprevalence of 59.1% for *E. canis* in dogs. The high prevalence of *E. canis* in dogs in areas where *R. linnaei* predominate is a common occurrence in Brazil [48]. In this regard, adults of *R. linnaei* are more efficient than nymphs for transmitting *E. canis* [49], but are rarely found parasitizing cats [4]. The lack of exposure of cats to *R. linnaei* adults and nymphs may be the reason for the absence of *E. canis*-positive cats in our study.

This work presents a main limitation, which is the fact that cats included in the analyses were referred for veterinary care, which could result in an overestimation of the CBC abnormalities. Indeed, CBC abnormalities in naturally infected cats should be interpreted with caution due to the possibility of comorbidities that may not have been deeply investigated herein. This is particularly true for leukocytosis, which is not a common feature of *Mycoplasma* spp. infection in cats.

## 5. Conclusions

This study reports the presence of *Cytauxzoon* spp. and *Mycoplasma* spp. in cats in Goiás, expanding the known geographical distribution of these FVBD agents in South America. We also found an association between *Mycoplasma* qPCR positivity and the simultaneous presence of thrombocytopenia and leukocytosis, indicating that the clinical significance of *Mycoplasma* spp. infection in Brazilian cats should be further explored. Eventually, *Mycoplasma* spp. infection should be considered as a differential diagnosis of thrombocytopenia and leukocytosis in otherwise apparently healthy cats.

## Figures and Tables

**Figure 1 microorganisms-12-02171-f001:**
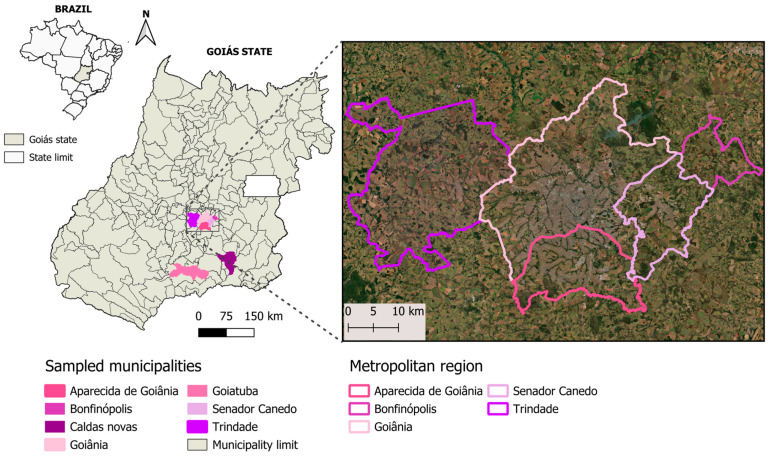
Urban locations in the state of Goiás, midwestern Brazil, where animals were sampled in the present study.

**Figure 2 microorganisms-12-02171-f002:**
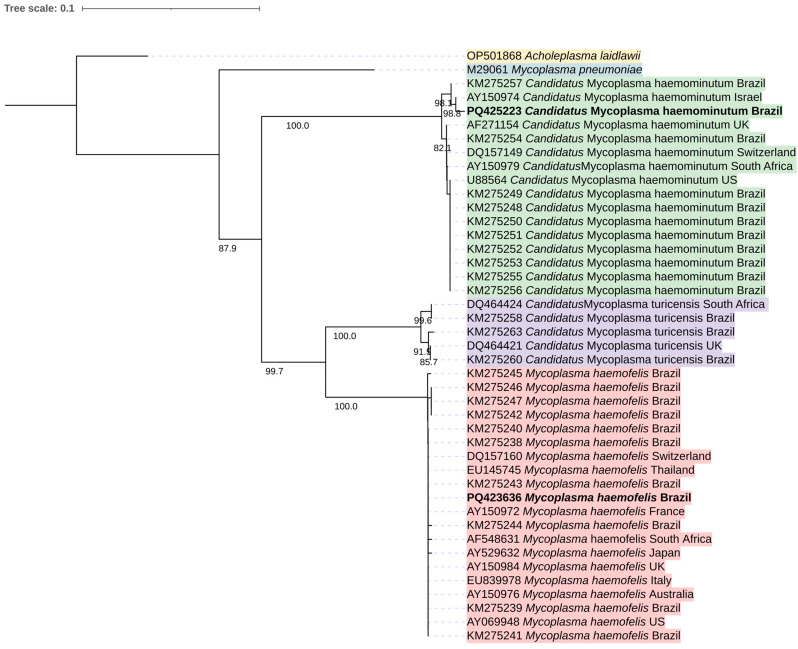
Phylogenetic tree reconstruction of the *Mycoplasma* genus based on a fragment of the *16S rRNA* gene using a dataset of 43 sequences (5 species). There were 524 positions in the final dataset. The evolutionary history was inferred using the neighbor-joining method and analyses were conducted in MEGA11. Bootstrap values < 70% are not shown. *Acholeplasma laidlawii* and *Mycoplasma pneumoniae* were used as outgroups. The final tree was edited in iTOL v.6.

**Table 2 microorganisms-12-02171-t002:** Classification of the cats according to age groups.

Age Groups	*n*	%
Kitten (<1 year)	6	4.4
Young adult (1–6 years)	54	40
Adult (7–9 years)	11	8.2
Senior (>9 years)	22	16.3
Unknown	42	31.1

*n* = number.

**Table 3 microorganisms-12-02171-t003:** *Cytauxzoon* and retrovirus positivity status in *Mycoplasma* spp.-positive cats.

Cat Number	*Mycoplasma* qPCR Positivity	*Cytauxzoon* cPCR Positivity	FIV Seropositivity	FELV Seropositivity
1	+	−	−	−
2	+	−	−	−
3	+	−	+	+
4	+	−	nt	nt
5	+	−	+	+
6	+	−	nt	nt
7	+	−	−	−
8	+	−	nt	nt
9	+	−	nt	nt
10	+	−	nt	nt
11	+	−	−	−
12	+	+	nt	nt
13	+	−	nt	nt
14	+	+	−	−
15	+	−	nt	nt
16	+	−	nt	nt
17	+	−	−	−
18	+	−	nt	nt
19	+	−	−	−
20	+	−	nt	nt
21	+	−	−	+
22	+	−	nt	nt
23	+	−	nt	nt
24	+	−	nt	nt
25	+	−	−	−
26	+	−	−	−
27	+	−	−	+
28	+	−	nt	nt

nt, not tested; +, positive; −, negative.

**Table 4 microorganisms-12-02171-t004:** Complete blood count (CBC) abnormalities in cats positive for *Mycoplasma* qPCR, feline retrovirus (FIV/FeLV) serology, or both.

CBC Abnormalities	*n*	%	*Mycoplasma*-Positive		FIV/FELV- Positive		*Mycoplasma*- and FIV/FELV-Positive	
			*n*	%	*n*	%	*n*	%
Anemia	21/135	15.5	6/21	28.6	4/21	19.0	1/21	4.8
Thrombocytopenia	71/135	52.6	19/71	26.8	6/71	8.4	3/71	4.2
Thrombocytosis	3/135	2.2	0/3	0	0/3	0	0/3	0
Leukopenia	11/135	8.1	1/11	9.1	0/11	0	0/11	0
Leukocytosis	30/135	22.2	10/30	33.3	1/30	3.3	1/30	3.3
Anemia + thrombocytopenia	10/135	7.4	4/10	40	3/10	30	1/10	10
Anemia + thrombocytosis	2/135	1.5	0/2	0	0/2	0	0/2	0
Anemia + leukopenia	2/135	1.5	0/2	0	0/2	0	0/2	0
Anemia + leukocytosis	5/135	3.7	3/5	60	1/5	20	1/5	20
Thrombocytopenia + leukopenia	3/135	2.2	0/3	0	0/3	0	0/3	0
Thrombocytopenia + leukocytosis	15/135	11.1	8/15	53.3	1/15	6.7	1/15	6.7
Thrombocytosis + leukopenia	1/135	0.7	0/1	0	0/1	0	0/1	0

*n* = number.

**Table 5 microorganisms-12-02171-t005:** Association between the most frequent complete blood count (CBC) abnormalities (anemia, thrombocytopenia, leukocytosis, and the combination of the last two) and *Mycoplasma* qPCR positivity in cats.

CBC Abnormalities	Present?	*Mycoplasma* qPCR		Chi-Squared Test	*p*-Value
		Positive	Negative		
Anemia	Yes	6	15	0.928	0.3355
	No	22	92		
Thrombocytopenia	Yes	19	52	3.301	0.0692
	No	9	55		
Leukocytosis	Yes	10	20	3.721	0.0537
	No	18	87		
Thrombocytopenia + leukocytosis	Yes	8	7	10.905	0.0010
	No	20	100		

## Data Availability

Data supporting the conclusions of this study are included in the manuscript. The *16S rRNA* sequences generated herein were deposited in GenBank (accession numbers: PQ423636; PQ425223).

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
