# Peer review of "Feline Vector-Borne Diseases and Their Possible Association with Hematological Abnormalities in Cats from Midwestern Brazil"

_microorganisms, 2024, doi:10.3390/microorganisms12112171_

Round 1

Reviewer 1 Report

Comments and Suggestions for Authors

Manuscript Title:  Feline Vector-Borne Diseases and Their Possible Association with Hematological Abnormalities in Cats From Midwestern Brazil.

Aim: This study aimed to detect Cytauxzoon spp., Ehrlichia spp., and Mycoplasma  spp. in cats referred to a teaching veterinary hospital in midwestern Brazil. In addition, the authors  investigated existing co-infections with Feline Immunodeficiency Virus (FIV) and Feline Leukemia Virus (FeLV) and assessed the correlation between Mycoplasma spp. positivity and cat variables, including age, sex, breed, and complete blood count (CBC) abnormalities.

General comments:

The study design is appropriate, the methods, results and discussion sections were clearly presented.

Specific comments:

However, the study title is interesting and presenting important data to the reader, the are some comments that should be corrected by the authors:

-            Please improve the English language of the manuscript.

-            Line 66: delete was

-            Line 114: please write full name for these abbreviations

-            Table 1: please write the name of genes in italic all over the manuscript

-            Line 148: please determine the number of samples sequenced

-            Line 229: please write small paragraph discuss the aim of this study

Comments on the Quality of English Language

-            Please improve the English language of the manuscript.

Author Response

Dear Reviewer 1, 

Please find the below responses to review comments:

Manuscript Title: Feline Vector-Borne Diseases and Their Possible Association with Hematological Abnormalities in Cats From Midwestern Brazil.
Aim: This study aimed to detect Cytauxzoon spp., Ehrlichia spp., and Mycoplasma spp. in cats referred to a teaching veterinary hospital in midwestern Brazil. In addition, the authors investigated existing co-infections with Feline Immunodeficiency Virus (FIV) and Feline Leukemia Virus (FeLV) and assessed the correlation between Mycoplasma spp. positivity and cat variables, including age, sex, breed, and complete blood count (CBC) abnormalities.

General comments:

The study design is appropriate, the methods, results and discussion sections were clearly presented.

Author’s response: We appreciate your comments and suggestions for improvements to the manuscript. All suggestions were accepted and addressed.

Specific comments:

However, the study title is interesting and presenting important data to the reader, the are some comments that should be corrected by the authors:
- Please improve the English language of the manuscript.

Author’s response: The English language of the manuscript has been revised.

- Line 66: delete was
Author’s response: Done.

- Line 114: please write full name for these abbreviations
Author’s response: Done.

- Table 1: please write the name of genes in italic all over the manuscript
Author’s response: Done.

- Line 148: please determine the number of samples sequenced
Author’s response: Done.

- Line 229: please write small paragraph discuss the aim of this study
Author’s response: We are very grateful for the suggestion. However, we would like to continue the discussion without this paragraph, since we have already included the objective of the study in the Abstract and in the last paragraph of the introduction:

"In this perspective, this molecular survey aimed to detect Cytauxzoon spp., Ehrlichia spp., and Mycoplasma spp. in cats referred to a teaching veterinary hospital in midwestern Brazil. We also investigated existing co-infections with Feline Immunodeficiency Virus (FIV) and Feline Leukemia Virus (FeLV) and assessed the correlation between Mycoplasma spp. positivity and cat variables, including age, sex, breed, and complete blood count (CBC) abnormalities."

Reviewer 2 Report

Comments and Suggestions for Authors

This is a nicely done study. The methods are well described, and the results are presented well. The authors have stated their conclusions carefully and with the study limitations in mind. I have only a few minor issues I would like the authors to address.

Line 90: How was randomization performed (e.g. method, software)? If selection was haphazard, please state so.

Line 98-101: Here it is stated that the hematocrit was performed with an automated cell counter. However, on line 104-105 hematocrit was determined by the microhematocrit method. Also note, that the microhematocrit method gives the packed cell volume (PCV) (https://eclinpath.com/hematology/tests/hematocrit/)

Line 106: Why were reference values from literature used? The automated counter used seems to be suitable for animals’ specimens and likely has animal reference intervals integrated. Literature reference values should always be a last resort.

Line 164: the beginning of this line should likely be in the figure text. It feel out of place here.

Line 182-184: I applaud the authors for including the test value and degrees of freedom, this is seldom seen. Well done!

Line 187-188: In line 141-142 it says that only specimens that were negative in for pathogens were further tested, but here it seems that all were tested. Please crosscheck.

Line 195-200: It is sad that only two sequences were identified. It would, however, be good if the authors could describe what the problem was with the other specimens? Did they suspect multiple species and therefore mixed sequences or something else?

Line 211-212: I think it would be good to remind the reader what these results mean, i.e. FeLV means antigen in the specimen and FIV means antibodies to the virus.

Table 5: Anemia has an extra line under it compared to the other

Author Response

Dear Reviewer 2, 

Please find the below responses to review comments:

This is a nicely done study. The methods are well described, and the results are presented well. The authors have stated their conclusions carefully and with the study limitations in mind. I have only a few minor issues I would like the authors to address.
Author’s response: We are grateful for your compliments and suggestions, which were accepted and addressed.

Line 90: How was randomization performed (e.g. method, software)? If selection was haphazard, please state so.
Author’s response: Revised as “Blood samples were collected (convenience sampling) between January 2022 and December 2023, as part of the routine veterinary care provided by the veterinary teaching hospital.”

Line 98-101: Here it is stated that the hematocrit was performed with an automated cell counter. However, on line 104-105 hematocrit was determined by the microhematocrit method. Also note, that the microhematocrit method gives the packed cell volume (PCV) (https://eclinpath.com/hematology/tests/hematocrit/)
Author’s response: Thank you for your comment. The correction has been made.

Line 106: Why were reference values from literature used? The automated counter used seems to be suitable for animals’ specimens and likely has animal reference intervals integrated. Literature reference values should always be a last resort.

Author’s response: The Celltac Alpha/MEK-6650Ⓡ (Nihon Kohden) automated cell analyzer has been previously validated for feline patient samples. To ensure the accuracy of the device, manual methods were employed during the validation process (McDaniel et al., 2013). For this specific study, samples from healthy cats in Goiânia and the surrounding region were analyzed, and the results were consistent with the reference values established in the literature (Weiss and Wardrop, 2011). Additionally, other studies that used samples analyzed by this device, where similar methodologies were applied, have already been published in scientific journals (Mendonça et al., 2024; Paula et al., 2022).

  • McDaniel, B.J.; Hirschberger, J; Weber, K. Validation of the Celltac alpha automated hematology analyzer for canine and feline blood samples. Veterinary Clinical Pathology 2013, 42 (1), 11-18, doi: 10.1111/vcp.12019.
  • Weiss, D.J.; Wardrop, K.J. Schalm’s Veterinary Hematology.; Wiley, 2011; ISBN 9780470961834.
  • de Mendonça, D. R.; Couto, L. F. M.; Pureza, L. H.; Martins, D. B.; Soares, V. E.; Ferreira, L. L.; Lopes, W. D. Z. First record of a possible trypanotolerant cattle breed in Latin America: Parasitological, serological, and clinical aspects. Veterinary Parasitology: Regional Studies and Reports 2024, 54, 101090, doi: 10.1016/j.vprsr.2024.101090.
  • Paula, W. V. de F., Taques, Í. I. G. G., Miranda, V. C., Barreto, A. L. G., Paula, L. G. F. de ., Martins, D. B., Damasceno, A. D., Muñoz-Leal, S., Sevá, A. da P., Dantas-Torres, F., Aguiar, D. M. de ., & Krawczak, F. da S.. (2022). Seroprevalence and hematological abnormalities associated with Ehrlichia canis in dogs referred to a veterinary teaching hospital in central-western Brazil. Ciência Rural, 52(2), e20201131. https://doi.org/10.1590/0103-8478cr20201131

Line 164: the beginning of this line should likely be in the figure text. It feel out of place here.
Author’s response: Thank you for your suggestion. We have checked the text and the information is in the correct place (Materials and Methods)

Line 182-184: I applaud the authors for including the test value and degrees of freedom, this is seldom seen. Well done!
Author’s response: Authors are deeply grateful for the compliments.

Line 187-188: In line 141-142 it says that only specimens that were negative in for pathogens were further tested, but here it seems that all were tested. Please crosscheck.
Author’s response: Revised as “All samples that were negative for Cytauxzoon spp., Ehrlichia spp., and Mycoplasma spp. tested positive in the cPCR assay for the detection of the mammalian endogenous control (cytB gene) and, therefore, were included in the analyses.”

Line 195-200: It is sad that only two sequences were identified. It would, however, be good if the authors could describe what the problem was with the other specimens? Did they suspect multiple species and therefore mixed sequences or something else?
Author’s response: The attempts to obtained additional sequences failed, probably due to the low amount of DNA, low quality DNA, or both. It now reads as follows: “The low number (only two) of sequences obtained can be attributed to low number of copies of the target DNA in the tested samples.”

Line 211-212: I think it would be good to remind the reader what these results mean, i.e. FeLV means antigen in the specimen and FIV means antibodies to the virus.
Author’s response: Thanks. We added the information: “This ELISA detects FeLV antigens and anti- FIV antibodies.”

Table 5: Anemia has an extra line under it compared to the other
Author’s response: Thank you very much for the correction. We fixed it.

Best regards,